# Low-Level Radon Activity Concentration—A MetroRADON International Intercomparison

**DOI:** 10.3390/ijerph19105810

**Published:** 2022-05-10

**Authors:** Petr P. S. Otahal, Eliska Fialova, Josef Vosahlik, Hannah Wiedner, Claudia Grossi, Arturo Vargas, Nathalie Michielsen, Tuukka Turtiainen, Aurelian Luca, Katarzyna Wołoszczuk, Thomas R. Beck

**Affiliations:** 1Nuclear Protection Department, National Institute for Nuclear, Chemical & Biological Protection, 26231 Milin, Czech Republic; fialovaeliska@sujchbo.cz (E.F.); vosahlik@sujchbo.cz (J.V.); 2Department of Geological Sciences, Faculty of Science, Masaryk University, 60200 Brno, Czech Republic; 3Physikalisch-Technischer Prüfdienst, Bundesamt für Eich- und Vermessungswesen, 1160 Wien, Austria; hannah.wiedner@wien.gv.at; 4Institut de Tècniques Energètiques, Universitat Politècnica de Catalunya, 08028 Barcelona, Spain; claudia.grossi@upc.edu (C.G.); arturo.vargas@upc.edu (A.V.); 5Institut de Radioprotection et de Sûreté Nucléaire, CEDEX, 92262 Fontenay-aux-Roses, France; nathalie.michielsen@irsn.fr; 6Radiation and Nuclear Safety Authority (STUK), Environmental Radiation Surveillance, 00811 Helsinki, Finland; tuukka.turtiainen@stuk.fi; 7Institutul National de Cercetare-Dezvoltare Pentru Fizica si Inginerie Nucleara “Horia Hulubei”, RO-077125 Magurele, Romania; aluca@nipne.ro; 8Central Laboratory for Radiological Protection, PL03194 Warsaw, Poland; woloszczuk@clor.waw.pl; 9Federal Office for Radiation Protection, 10318 Berlin, Germany; tbeck@bfs.de

**Keywords:** radon, interlaboratory comparison, radon activity concentration, calibration, traceability

## Abstract

An international comparison of continuous monitors measuring radon activity concentration was performed to validate the traceability of the European radon calibration facilities. It was carried out by comparing the secondary standards used by these previous facilities, ranging from 100 Bq·m^−3^ to 300 Bq·m^−3^. Secondary standards were individually compared to a secondary reference device previously calibrated in a reference radon atmosphere traceable to a primary standard. The intercomparison was organized by the National Institute for Nuclear, Chemical, and Biological Protection (SUJCHBO) in the period from October 2019 to April 2020 within the European Metrology Program for Innovation and Research (EMPIR), JRP-Contract 16ENV10 MetroRADON. Eight European laboratories participated in this study. The results of the experiment are presented and discussed.

## 1. Introduction

Radon is a colourless noble gas without taste and odour and is a naturally radioactive element. It occurs in various quantities as an intermediate step in the radioactive decay chains, through which thorium and uranium slowly decay into lead and other short-lived radioactive elements. Since thorium and uranium are two of the most common radioactive elements on Earth, also having three isotopes with half-lives on the order of several billion years, radon will be present on Earth long into the future despite its relatively short half-life (3.8232 ± 0.0008 days, with the uncertainty corresponding to a coverage factor k = 1) [1].

Radon can be present in the environment in different concentrations depending on the geological character of the area. Radon activity concentrations ordinarily measured in soil could vary in the range from tens to hundreds of kilobecquerels per cubic meter. On the other side, after being exhaled from the ground to the atmosphere, radon is diluted within the atmospheric boundary layer. Its atmospheric concentrations are usually in the range of tens of becquerels per cubic meter [2,3]. However, the radon activity concentrations in some public accessible subterranean spaces (for example, tourist caves, speleotherapy caves or underground tours built up from former mining workplaces) could be measured in the range of hundreds becquerels per cubic meterup to tens of kilobecquerels per cubic meter. Radon activity concentrations in indoor air of residential or long-stay rooms (for example, dwellings, workplaces, educational or social facilities) are usually not higher than a few hundreds of becquerels per cubic meter. However, in unfavourable conditions, depending on the geological background and indoor air-exchange ventilation conditions, the radon activity concentrations in indoor air could exceed the values of several kilobecquerels per cubic meter. More detailed information about the general properties of radon can be found in the European Atlas of Natural Radiation [1].

The European Council Directive 2013/59/EURATOM (EU-BSS) [4] brings new challenges for European laboratories in the metrology and calibration of devices measuring radon. According to the EU-Basic Safety Standards, the levels of relevant activity concentration should not exceed 300 Bq·m^−3^. The historical development and acceptance of the EU documents by the member states place much higher demands for the measurement’s precision. This is the main reason for an unceasing effort of commercial radon monitors´ calibration procedures development.

The radon activity concentration´s measurement precision ensures an adequate assessment of the occupational and general public exposure to radon and its decay products. The significance of accurate measurement of radon concentration is also gaining importance concerning the new conversion coefficients presented in ICRP 137 [5]. For precise measurement below 300 Bq·m^−3^, it is necessary to quantitatively evaluate the accuracy of different methods and equipment used to measure the radon activity concentration.

The National Institute for Nuclear, Chemical and Biological Protection (SUJCHBO) has been operating the Authorized Metrological Center since 1991. This centre ensures the metrological traceability for devices measuring the radon activity concentration and the equilibrium equivalent radon activity concentration for the Czech Republic. Primary measuring devices are regularly subject to metrological traceability at leading European laboratories. The Authorized Metrological Center working under SUJCHBO is authorized by the Czech office for standards, metrology and testing, accredited by the Czech Institute for Accreditation according to the standard ČSN EN ISO/IEC 17025:2018 and certificated by Lloyd´s register quality insurance.

The metrological testing and calibration of radon monitors require a long-term stable radon atmosphere. SUJCHBO, as a part of the MetroRADON project solution, has developed a unique device for the calibration of measuring instruments at low-level radon activity concentrations (low-level radon chamber, LLRCH) below 300 Bq·m^−3^. During the process of the radon chamber fabrication, many experiments that demand the adjustment of various low-level radon activity concentrations were provided. The airtightness and sustainability of long-term stable radon atmospheres in the LLRCH were verified by them. On the grounds of these results, SUJCHBO was able to organize an on-site comparison measurement to verify the traceability of secondary standards used by leading radon calibration laboratories in Europe. The task aimed at the calibration of secondary standards of European calibration laboratories was performed in the framework of the MetroRadon project in the period from October 2019 to April 2020 in the SUJCHBO´s laboratory. The secondary purpose of the intercomparison, in addition to the calibration of the different radon activity concentration measuring devices from significant European laboratories, was to disseminate the best laboratory practice to the professional public.

## 2. Organization and Methodology

In order to achieve a specific low-level radon activity concentration, it is necessary to ensure a constant radon supply and defined ventilation in the radon chamber. As a radon source, we used a flow-through source developed by CMI (Czech Metrological Institute) and designed based on F. Kysela’s instructions (from the Institute for Research, Production, and Utilization of Radioisotopes in Prague, Czech Republic). The whole system of radon activity time-stable atmosphere is operated based on the model of constant input and constant ventilation conditions:(1)$$a_{t}=a_{o\;}\cdot \;e^{-\left(\lambda_{\mathrm{Rn}}+k\right).t}+\frac{\varrho }{V\left(k+\lambda_{\mathrm{Rn}}\right)}{\left(1-e^{-\left(\lambda_{Rn}+k\right).t}\right)}_{}$$
$a_{t}$volume activity at time t (Bq⋅m^−3^)$a_{o\;}$volume activity at time 0 (Bq⋅m^−3^)λ_Rn_radon decay rate constant (h^−1^)kmultiple air exchange (h^−1^)ttime (h)ϱradon feed rate (Bq⋅h^−1^)Vvolume of radon chamber (m^3^)

For the steady state (t = ∞) at constant air exchange multiplicity and constant radon feed rate, the following applies:(2)$$a_{V,\mathrm{Rn}}=R_{\mathrm{Rn}}\;/(Q\;\;\;\frac{M.p\;\mathrm{at}\;Q\;\mathrm{calibration}\;}{R.T\;\mathrm{at}\;Q\;\mathrm{calibration}}/\;\frac{M.p\;\mathrm{at}\;\mathrm{Rn}\;\mathrm{confrontation}\;}{R.T\;\mathrm{at}\;\mathrm{Rn}\;\mathrm{confrontation}}+\lambda_{\mathrm{Rn}}.V)$$
a_V,Rn_radon activity concentration (Bq⋅m^−3^)Qflow rate (m^3^⋅h^−1^)Mmolar mass (kg⋅mol^−1^)p _at Q calibration_
air pressure 1013,25 (hPa)Rmolar gas constant (J⋅mol^−1^⋅K^−1^)T _at Q calibration_
temperature 273,16 (K)p_at Rn confrontation_air pressure (Pa)T_at Rn confrontation_
temperature (K)λ_Rn_radon decay rate constant (h^−1^)Vvolume of radon chamber (m^3^)R_Rn_
radon emanation power (Bq⋅h^−1^)

For the radon monitors comparison measurement, we used an infrastructure which enabled us to maintain the long-term stable radon atmosphere. In particular, the equipment consisted of an airtight low-level radon chamber (LLRCH), a humidifier, a flow-through source of Rn-222 type RF 5 (Prague, Czech Republic), a mass flow controller of the Bronkhorst^®^ EL-Flow type (Bethlehem, PA, USA), an aerosol filter and an air pressure vessel as the source of radon-free air [6].

As reported in Table 1, eight European calibration laboratories participated in the intercomparison—BEV-PTP (Austria), INTE-UPC (Spain), IRSN (France), STUK (Finland), IFIN-HH (Romania), CLOR (Poland), BfS (Germany) and SUJCHBO (Czech Republic). The participants´ instruments were individually sent to the SUJCHBO laboratory according to a prearranged schedule and were calibrated at two different levels of stable radon activity concentration (200 Bq·m^−3^ and 300 Bq·m^−3^) [7]. In addition, each monitor´s background was determined in an atmosphere with zero radon activity concentration.

AlphaGuard PQ2000 (serial number 1001) was selected as SUJCHBO’s reference measuring instrument (RMI). This device was calibrated by BfS Berlin in March 2019 (calibration certificate no. R-19-1) and by PTB Braunschweig in November 2012 (calibration certificate no. PTB-6.13-77-Rn222-S). RMI was always put in the LLRCH together with the participant’s individual calibrated radon monitor. The main technical parameters of the RMI during each comparison measurement are summarized in Table 2.

The exposure time represents the time-stable radon activity concentration at the required level. According to the EA 04/02 document [8], an expanded uncertainty of 2% for 100 Bq·m^−3^ of the time-stable radon atmosphere was derived.

For the evaluation of the secondary standards calibration results, the document ISO 13528:2015 “Statistical methods for use in proficiency testing by interlaboratory comparison” [9] was used. The z-score calculated using the participants´ results, assigned value, and the standard deviation for proficiency assessment was used.

After the calibration process was finished, the participant´s device was sent back to download data and process results. Participants forwarded to the coordinator their measurement results—the average of radon activity concentration in the exposure period which was defined by SUJCHBO. Based on data provided by the participants, the values of the z-score and deviation estimation for each level of radon activity concentration were calculated.
Determination of Z Score:
(3)$$Z=\frac{Xi-Xpt}{\delta \mathrm{pt}}$$
where Xi is the participant’s result; Xpt is the reference value; δpt is the standard deviation (in this case 10%) [9].

The standard deviation level of 10% was determined based on experiences with previous rounds of proficiency testing for the same measurement with comparable property values and by using compatible measurement procedures by participants [7].
Interpretation of Z Score
|Z|≤2.0the result is considered acceptable2.0<|Z|<3.0the result is considered a source of warning|Z|≥3.0the result is considered unacceptable
Deviation Estimation (Measurement Error):
(4)$$\;Di\%=100\;\frac{\left(Xi-Xpt\right)}{Xpt}$$
where Xi is the participant’s result; Xpt is the reference value

## 3. Results and Discussion

### 3.1. Background Activity Concentration

All measuring devices were placed in the LLRCH with the zero-radon activity concentration. The resulting value of the background activity was calculated over 24 h of exposition in the atmosphere without radon. The uncertainty of participant measurement (after the exposition in the LLRCH, participant received their device back, downloaded the data and sent their results to SUJCHBO) is an expanded uncertainty, as the product of the standard measurement uncertainty and the coverage factor k = 2 indicates approximate 95% confidence. The deviations of participants nos. 2, 3 and 8 were very low, so they cannot be shown in the plot. The reader should take into account that the participants order is not the same as in Table 1 (measurements are reported in random order).

The background activity of radon monitors originates mainly from ^210^Pb (respectively, ^210^Po) contamination and from electrical noise. The contamination of detectors increases continuously throughout the instrument’s life, depending on the length of use and the concentrations of radon in which the instrument measures and is usually stored. Some devices allow an automatic subtraction of the background value. However, most measuring devices do not allow automatic background subtracting. Thus, at low values of radon activity concentration, there may be an inadequate increase of measured values (see Figure 1). From measurements in the zero-radon-activity concentration, the partners found the correction factor for the background subtraction for the next part of the intercomparison.

### 3.2. Participants’ Results for a Level of 200 Bq·m^−3^

The results given in Table 3 and Figure 2 represent the average values from the individual measurements. The reference level of the average radon activity concentration was calculated based on the results of the calibrated reference measuring instrument AlphaGuard PQ2000.

### 3.3. Participants’ Results for a Level of 300 Bq·m^−3^

The results given in Table 4 and Figure 3 represent the average values from the individual measurements. The reference level of average radon activity concentration was calculated based on the results of the calibrated reference measuring instrument AlphaGuard PQ2000.

During the intercomparison, it was necessary to use more than 30 m^3^ of the radon free air for one measuring campaign (one participant´s device). The whole calibration campaign, including the administration associated with the device’s transport, calibration under two levels of radon activity concentration and determination of the device´s background, took not more than two weeks.

## 4. Conclusions

The calibration of the European calibration laboratories´ secondary standards was performed in the period from October 2019 to April 2020. Eight leading European laboratories attended the intercomparison, which was organized and performed by the responsible staff of SUJCHBO. The singular equipment called the low-level radon chamber developed within the MetroRADON project framework was used to test measuring devices in a low-level radon atmosphere. The comparison measurement campaign was realized at two different levels of radon activity concentration, at 200 Bq·m^−3^ and 300 Bq·m^−3^. On top of that, all devices’ background activities were determined during the intercomparison measurement process.

The main parameters for the evaluation of the participants’ results were the z-score and the estimation of the standard deviation.

For a level of 200 Bq·m^−3^, the values of the z-score varied from −0.6 up to 0.8 and the values of the deviation estimation ranged from −3.0% up to 4.0%.

For a level of 300 Bq·m^−3^, the values of the z-score varied from −1.2 up to 0.6 and the values of the deviation estimation ranged from −4.0% up to 2.0%.

Based on the appraisal of individual participants’ results parameters (maximum z-score and deviation D), it was possible to pronounce that the results of the European calibration laboratories’ secondary standards were at the exact level.

Upon successful completion, all intercomparison participants received an official accredited calibration certificate, demonstrating the metrological traceability between the individual calibration and measuring laboratories within the EU.

The intercomparison measurement of European calibration laboratories´ secondary standards for radon calibration is an effective tool for detecting discrepancies in metrological traceability. It validates the high-quality of radon measurements in Europe. It is strongly advised to perform the calibration and verification of devices on a regular basis.

## Figures and Tables

**Figure 1 ijerph-19-05810-f001:**
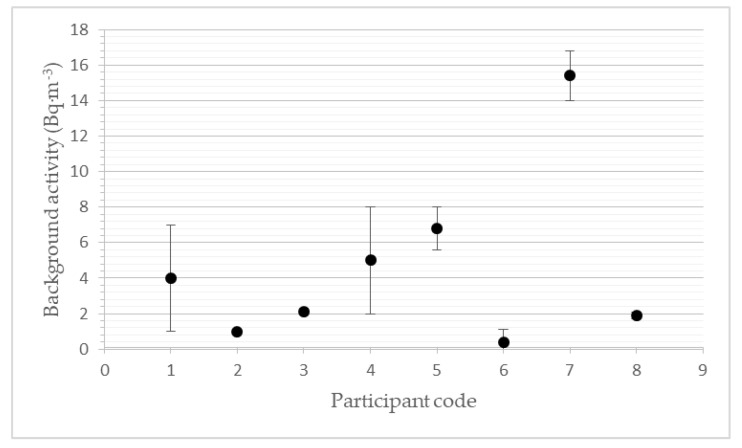
Results of participants’ devices in a zero-radon-activity concentration atmosphere.

**Figure 2 ijerph-19-05810-f002:**
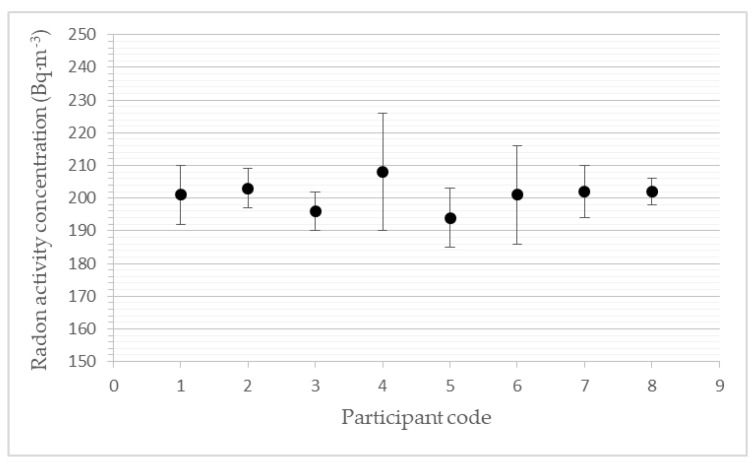
Participants’ devices results for the level of 200 Bq·m^−3^.

**Figure 3 ijerph-19-05810-f003:**
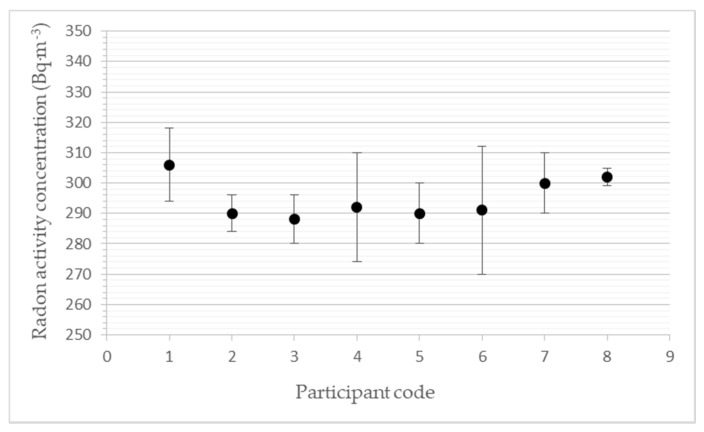
Participants’ results for a level of 300 Bq·m^−3^.

**Table 1 ijerph-19-05810-t001:** Radon calibration facilities participating in the interlaboratory comparison (sorted alphabetically by acronym) [7].

Acronym	Institute	Country	Measuring Device
BEV-PTP	Physikalisch-technischer Prüfdienst, Bundesamt für Eich- und Vermessungswesen, Wien	Austria	AG * PQ2000 Pro
BfS	Federal Office for Radiation Protection, Berlin	Germany	AG DF2000
CLOR	Central Laboratory for Radiological Protection, Warsaw	Poland	AG DF2000
IFIN-HH	Horia Hulubei National Institute for R&D in Physics and Nuclear Engineering, Bucharest	Romania	RadonScout
IRSN	Institut de Radioprotection et de Sûreté Nucléaire, Fontenay-aux-Roses	France	AG PQ2000
STUK	Radiation and Nuclear Safety Authority, Helsinki	Finland	AG PQ2000
SUJCHBO (Coordinator)	National Institute for Nuclear, Chemical and Biological Protection, Milin	Czech Republic	AG DF2000
INTE-UPC	Laboratory of ^222^Rn studies of the Institut de Tecniques Energetiques of the Universitat Politecnica de Catalunya, Barcelona	Spain	AG PQ2000 Pro

* AG: AlphaGuard.

**Table 2 ijerph-19-05810-t002:** Technical parameters of the intercomparison.

Parameter	Value
Integration time	60 min
Exposure time	at least 24 h
Temperature	22.3 ± 1.1 °C
Relative humidity	51.2 ± 3.8%
Air pressure	955.8 ± 8.6 hPa

**Table 3 ijerph-19-05810-t003:** Results for a level of radon activity concentration of 200 Bq·m^−3^.

Participants’ Code	a_v,Rn_ (Bq·m^−3^)	s(a_v,Rn_) (Bq·m^−3^)	z-Score (-)	D (%)
1	201	9	0.1	0.5
2	203	6	0.3	1.5
3	196	6	−0.4	−2.0
4	208	18	0.8	4.0
5	194	9	−0.6	−3.0
6	201	15	0.1	0.5
7	202	8	0.2	1.0
8	202	4	0.2	1.0

a_v,Rn_: the average participant´s value of radon activity concentration in the exposure period (each participant sent their results to SUJCHBO after they received the monitor back and downloaded the data). s(a_v,Rn_): participant´s measurement uncertainties (each participant sent his results to SUJCHBO after they received the monitor back and downloaded the data)—it is an expanded uncertainty—the product of the standard measurement uncertainty and the coverage factor k = 2.

**Table 4 ijerph-19-05810-t004:** Results for a level of radon activity concentration of 300 Bq·m^−3^.

Participants’ Code	a_v,Rn_ (Bq·m^−3^)	s(a_v,Rn_) (Bq·m^−3^)	z-Score (-)	D (%)
1	306	12	0.6	2.0
2	290	6	−1.0	−3.3
3	288	8	−1.2	−4.0
4	292	18	−0.8	−2.7
5	290	10	−1.0	−3.3
6	291	21	−0.9	−3.0
7	300	10	0	0
8	302	3	0.2	0.7

a_v,Rn_: the average participant´s value of radon activity concentration in the exposure period (each participant sent their results to SUJCHBO after they received the monitor back and downloaded the data). s(a_v,Rn_): participant´s measurement uncertainties (each participant sent their results to SUJCHBO after they received the monitor back and downloaded the data)—it is an expanded uncertainty—the product of the standard measurement uncertainty and the coverage factor k = 2.

## Data Availability

The data set as well as further analyses are available online at http://metroradon.eu/wp-content/uploads/2017/06/16ENV10-MetroRADON-D7-final accepted.pdf (accessed on 27 March 2022).

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
