# Peer review of "Low-Level Radon Activity Concentration—A MetroRADON International Intercomparison"

_ijerph, 2022, doi:10.3390/ijerph19105810_

Round 1
Reviewer 1 Report
The paper presents an inter-comparison of radon concentration measurement devices done at the National Institute for Nuclear, Chemical, and Bio-21 logical Protection (SUJCHBO) in the frame of the MetroRADON project.
The measurements show that all devices from partner institutes gave results consistent enough to allow their calibration accreditation.
The paper is well written and is of great interest for the community.
Author Response
Thank you for the time you spent by reviewing of the paper.
There are no suggestions for corrections.
Reviewer 2 Report
The paper reports an interesting comparison of several radon measurements from different European facilities to evaluate their compatibility, relative calibration and aimed to their final validation. The method is clear and rigorous and the results presented in the paper are valuable for exchange and interoperability of such measurements in the field of environmental researches.
The paper deserves publication in the journal provided a few minor fixes are taken into consideration.
Concerning the method, a couple of clarifications could be beneficial to the text:
- line 166: you indicate "coverage factor k=2". May you please explain in short, for the sake of completeness, what do you intend for
coverage factor? You mention it also in line 36, a short explanation here could be useful. - l 172-178: you state that some of the detectors provide an automatic background suppression (but we don't know which).
Does Figure 1 report both suppressed and not-suppressed results?
Does this suggest that the smallest values in Fig. 1 could be due to this correction, while the highest could bear an excess due to the
suppression not being performed? The text needs some proper clarification/rephrasing.
From the formal point of view:
- line 16: intercomparison: just use "comparison" ("inter" is implicit in the comparison concept)
- l. 31: the lead -> remove "the"
- l. 33: while also -> remove "while"
- l. 44: m-3 -> force -3 on the same line
- l. 47: the first hundreds -> a few hundreds
- l. 52: laid down -> fixed (?)
- l. 74: m-3 -> -3 as superscript
- l. 89: comes from -> remove comes
- table 1: please don't split the table across pages
- l. 167: Please... Table 1 -> do you mean that the measurements are reported in random order not to trace the facility which reported them? Probably you should rephrase the sentence for better clarity.
- l. 172: (respectively 210Po) -> or 210Po
- table 3: as above, please don't split the table across pages. This is particularly important when results are reported, to provide an overall
estimation of results' quality by the reader.
Author Response
Thank you very much for your comments and time you spent with review of our paper. The corrections based on your recommendations are mentioned under each of your comments in bold.
Concerning the method, a couple of clarifications could be beneficial to the text:
line 166: you indicate "coverage factor k=2". May you please explain in short, for the sake of completeness, what do you intend for
coverage factor? You mention it also in line 36, a short explanation here could be useful.
Explained in line 181: “indicates approximate 95% confidence”.
l 172-178: you state that some of the detectors provide an automatic background suppression (but we don't know which).
It was explained in lines 195 and 196: “From measurements in the zero radon activity concentration, the partners found the correction factor for the background subtraction for the next part of the intercomparison”.
Does Figure 1 report both suppressed and not-suppressed results?
The figure 1 represents backgraund results of each participant´s device. It is not possible to substract background from these results. The sufficient explanation is mentioned in the text under the figure 1.
Does this suggest that the smallest values in Fig. 1 could be due to this correction, while the highest could bear an excess due to the
suppression not being performed? The text needs some proper clarification/rephrasing.
It was explained in the text.
From the formal point of view:
line 16: intercomparison: just use "comparison" ("inter" is implicit in the comparison concept)
AGREED
- 31: the lead -> remove "the"
AGREED
- 33: while also -> remove "while"
AGREED
- 44: m-3 -> force -3 on the same line
AGREED
- 47: the first hundreds -> a few hundreds
AGREED
- 52: laid down -> fixed (?)
The expression “laid down” is correct, it was preserved in the text.
- 74: m-3 -> -3 as superscript
AGREED
- 89: comes from -> remove comes
AGREED
table 1: please don't split the table across pages
AGREED
- 167: Please... Table 1 -> do you mean that the measurements are reported in random order not to trace the facility which reported them? Probably you should rephrase the sentence for better clarity.
This comment is explained in lines 183-185: “Please, the reader should take into account that participants 183 order are not numbered in agreement with Table 1 (measurements are reported in random 184 order).“ I think there is nothing for rephrasing.
- 172: (respectively 210Po) -> or 210Po
The expression “respectively” is correct, it was preserved in the text.
table 3: as above, please don't split the table across pages. This is particularly important when results are reported, to provide an overall estimation of results' quality by the reader.
AGREED
Reviewer 3 Report
The article bring results of a intercomparison exercise on relatively low radon concentrations. The need for such exercise was a result of introduction of the EC Directive from 2013, which introduced newer and lower maximum levels for radon-222 in human close environment. This context makes the paper very interesting and I recommend it for publication however after minor improvements, suggested below. It is very interesting finding that measurements at level of 200 Bq/m3 are not causing problems (all the results are well comparable) the low concentration produces an order of magnitude scattering of results.
Row 38. The abbreviation „NCA” should be resolved at first use . Is it “Non carrier added?”
Row 59,60. Simialar for “VWR”. It is not clear at all what it is.
Rows-38-49 – In such wide review of results it would be nice to have some references, for example to a review article or even UNSCEAR report.
Equation 2 (in fact it is Eq. 1) & row 95. The symbol used for decay constant of radon looks differently in equation (“Rn” is bold) and in explanation (nothing is bold). Moreover I think that writing “Rn” in the same level as lambda suggest rather some product of two quantities then a single quantity. It would be much better if the RN will be in lower index. However, since there is no other decay constant in this equation why do not use only lambda and do not write “Rn” at all.
Row 101 – 102 – this is the “real” equation 2 I think.
Why you use the same character “R” (once with index, the second time plain) to describe completely different quantities like radon emanation power and molar gas constant. There is plenty of character in Latin and Greek alphabets to choose some unique. Again I do not understand the need of adding index to Q when it is the only one quantity attributed to this character.
Altogether I will strongly recommend to use as simple symbols as possible in Equations, not to mix symbols and description in a formula.
Author Response
Thank you for the time you spent by reviewing of the paper. The corrections based on your recommendations are mentioned under each of your comments in bold.
Row 38. The abbreviation „NCA” should be resolved at first use . Is it “Non carrier added?”
The abbreviation “NCA” is not mentioned in the text, so it could not be reviewed.
Row 59,60. Simialar for “VWR”. It is not clear at all what it is.
The abbreviation “VWR” is not mentioned in the text, so it could not be reviewed.
Rows-38-49 – In such wide review of results it would be nice to have some references, for example to a review article or even UNSCEAR report.
In lines 49-50, new reference No. 4 was added (European Atlas of Natural Radiation).
Equation 2 (in fact it is Eq. 1) & row 95. The symbol used for decay constant of radon looks differently in equation (“Rn” is bold) and in explanation (nothing is bold). Moreover I think that writing “Rn” in the same level as lambda suggest rather some product of two quantities then a single quantity. It would be much better if the RN will be in lower index. However, since there is no other decay constant in this equation why do not use only lambda and do not write “Rn” at all.
It was agreed and repaired in equation 1.
Row 101 – 102 – this is the “real” equation 2 I think.
It was agreed and repaired in equation 2.
Why you use the same character “R” (once with index, the second time plain) to describe completely different quantities like radon emanation power and molar gas constant. There is plenty of character in Latin and Greek alphabets to choose some unique. Again I do not understand the need of adding index to Q when it is the only one quantity attributed to this character.
Altogether I will strongly recommend to use as simple symbols as possible in Equations, not to mix symbols and description in a formula.
All characters were checked and they are correctly explained in the explanatiory notes. The used characters were preserved in the equations.
This manuscript is a resubmission of an earlier submission. The following is a list of the peer review reports and author responses from that submission.